# A Multidisciplinary Approach to Fungal Infections: One-Year Experiences of a Center of Expertise in Mycology

**DOI:** 10.3390/jof6040274

**Published:** 2020-11-10

**Authors:** Nico A. F. Janssen, Roger J. M. Brüggemann, Monique H. Reijers, Stefanie S. V. Henriet, Jaap ten Oever, Quirijn de Mast, Yvonne Berk, Elizabeth A. de Kort, Bart Jan Kullberg, Mihai G. Netea, Jochem B. Buil, Janette C. Rahamat-Langendoen, Didi Bury, Eline W. Muilwijk, Jacques F. Meis, Paul E. Verweij, Frank L. van de Veerdonk

**Affiliations:** 1Center of Expertise in Mycology Radboudumc/CWZ, 6525 GA Nijmegen, The Netherlands; nico.janssen@radboudumc.nl (N.A.F.J.); roger.bruggemann@radboudumc.nl (R.J.M.B.); monique.reijers@radboudumc.nl (M.H.R.); stefanie.henriet@radboudumc.nl (S.S.V.H.); elizabeth.dekort@radboudumc.nl (E.A.d.K.); bj.kullberg@radboudumc.nl (B.J.K.); mihai.netea@radboudumc.nl (M.G.N.); jochem.buil@radboudumc.nl (J.B.B.); j.meis@cwz.nl (J.F.M.); paul.verweij@radboudumc.nl (P.E.V.); 2Department of Internal Medicine, Radboud University Medical Center, 6525 GA Nijmegen, The Netherlands; jaap.tenoever@radboudumc.nl (J.t.O.); quirijn.demast@radboudumc.nl (Q.d.M.); 3Department of Pharmacy, Radboud University Medical Center, 6525 GA Nijmegen, The Netherlands; didi.bury@radboudumc.nl (D.B.); e.w.muilwijk@prinsesmaximacentrum.nl (E.W.M.); 4Department of Pulmonology, Radboud University Medical Center, 6525 GA Nijmegen, The Netherlands; 5Department of Pediatric Infectious Diseases and Immunology, Radboudumc Amalia Children’s Hospital, Radboud University Medical Center, 6525 GA Nijmegen, The Netherlands; 6Department of Pulmonary Diseases, Canisius-Wilhelmina Hospital (CWZ), 6532 SZ Nijmegen, The Netherlands; y.berk@cwz.nl; 7Department of Hematology, Radboud University Medical Center, 6525 GA Nijmegen, The Netherlands; 8Department of Medical Microbiology, Radboud University Medical Center, 6525 GA Nijmegen, The Netherlands; janette.rahamat-langendoen@radboudumc.nl; 9Department of Medical Microbiology and Infectious Diseases, Canisius-Wilhelmina Hospital (CWZ), 6532 SZ Nijmegen, The Netherlands

**Keywords:** mycology, invasive fungal diseases, multidisciplinary, diagnosis, antifungal treatment

## Abstract

Invasive fungal diseases (IFDs) often represent complicated infections in complex patient populations. The Center of Expertise in Mycology Radboudumc/CWZ (EMRC) organizes a biweekly multidisciplinary mycology meeting to discuss patients with severe fungal infections and to provide comprehensive advice regarding diagnosis and treatment. Here, we describe the patient population discussed at these meetings during a one-year period with regards to their past medical history, diagnosis, microbiological and other diagnostic test results and antifungal therapy. The majority of patients discussed were adults (83.1%), 62.5% of whom suffered from pulmonary infections or signs/symptoms, 10.9% from otorhinolaryngeal infections and/or oesophagitis, 9.4% from systemic infections and 9.4% from central nervous system infections. Among children, 53.8% had pulmonary infections or signs/symptoms, 23.1% systemic fungal infections and 23.1% other, miscellaneous fungal infections. 52.5% of adult patients with pulmonary infections/symptoms fulfilled diagnostic criteria for chronic pulmonary aspergillosis (CPA). Culture or polymerase chain reaction (PCR) demonstrated fungal pathogens in 81.8% of patients, most commonly *Aspergillus*. A multidisciplinary mycology meeting can be a useful addition to the care for patients with (I)FDs and can potentially aid in identifying healthcare and research needs regarding the field of fungal infections. The majority of patients discussed at the multidisciplinary meetings suffered from pulmonary infections, predominantly CPA.

## 1. Introduction

Fungal infections have a large impact on human health, and invasive fungal diseases (IFDs) are associated with high mortality rates [1,2,3]. It is estimated that global mortality due to IFDs equals or surpasses that of tuberculosis (TBC) or malaria [1]. The reported changing epidemiology of IFDs in human hosts [4,5] is partly due to a growing immunocompromised patient population at risk of these infections, such as those using immunosuppressive medication, allogeneic stem cell and/or solid organ transplant recipients, and those suffering from human immunodeficiency virus (HIV) infection or acquired immunodeficiency syndrome (AIDS) [1]. Furthermore, new clinical IFD entities have been recognized, such as invasive pulmonary aspergillosis (IPA) occurring in severely ill influenza patients, with a high incidence in some geographic regions and a high mortality rate [6,7,8], and coronavirus disease 2019 (COVID-19)-associated pulmonary aspergillosis [9,10]. In addition, antifungal resistance has been documented in multiple countries [11], is increasing [12,13], and directly impacts patient survival. Overall mortality rates are significantly higher in patients with culture-positive voriconazole-resistant invasive aspergillosis (IA), compared to voriconazole-susceptible cases [14,15]. Last, the number of antifungal drug classes is limited, and current antifungal agents have many drug interactions with concomitant medication. These developments have increased the need for expertise on fungal pathogens, diagnostic options, and resistance management (medical mycology), specific immune defects in (I)FD patients and options for immunotherapy (adult/paediatric infectious diseases), and antifungal drugs, with the required exposure profile and drug-interaction management (clinical pharmacology). As experts in each of these three areas were increasingly consulted individually regarding patients with complex fungal diseases, we decided to combine their complementary expertise in the Center of Expertise in Mycology Radboudumc/CWZ (EMRC), and to set up a multidisciplinary mycology meeting, open to other hospitals to discuss cases through virtual conferencing (Figure 1). A pulmonologist and haematologist joined our expert team to provide additional expertise in specific patient populations, comparable to the previously described multidisciplinary team for antifungal stewardship [16].

During these meetings, patient cases are discussed in a structured manner regarding medical history, clinical courses, diagnostic test results, microbiological aspects, host immunological aspects, pharmacological issues, and treatment provided so far. Subsequently, recommendations are formulated regarding further diagnostic work-up, treatment, and/or follow-up. Additionally, further specialized microbiological tests (e.g., antifungal susceptibility testing), translational immunological work-up (e.g., cytokine production assays), and pharmacological laboratory testing (e.g., therapeutic drug monitoring (TDM)) are offered, if necessary. Currently, the EMRC is not involved in antifungal stewardship activities, but offers advice on antifungal treatment on a case-by-case basis. Recommendations for antifungal stewardship have recently been published [17]. Patients with fungal diseases can be registered for the meeting by any physician involved in their care. By providing multidisciplinary expertise on fungal infections and offering specialized testing, the EMRC aims to positively impact patient care and translational research with clinical significance for the patient. As such, it offers a consultancy function accessible to physicians from across the world.

Here, we aim to provide an overview of the fungal infections and the clinical, microbiological and treatment issues encountered during the multidisciplinary mycology meetings that took place during the first year of systematic registration of the proceedings and outcomes of these meetings. Furthermore, we aim to provide others with a framework for establishing a multidisciplinary team to address the increasing complexity of fungal infection management.

## 2. Materials and Methods

All patients discussed during the biweekly multidisciplinary mycology meeting in the EMRC were registered in a database (Microsoft Excel). Patient history and clinical course (demographics, medical history, use of relevant medication, signs and symptoms), diagnostic test results (radiological examinations, relevant laboratory results), microbiological results (cultures, polymerase chain reaction (PCR) results, susceptibility testing), information on host immunological factors and employed antifungal treatment (including TDM results) were provided in advance by the presenting physician. If the patient primarily received treatment at Radboudumc, this information was extracted from the electronic patient record (Epic Hyperspace; Epic Systems Corporation, Verona, WI, USA).

For this study, data from patients discussed at the multidisciplinary meetings between 01-07-2017 and 30-06-2018 were included. Statistical analysis was performed using Microsoft Excel 2016 for Office 365 and GraphPad Prism 5 for Windows (GraphPad Software, Inc., San Diego, CA, USA).

## 3. Results

### 3.1. Meeting and Patient Characteristics

During the 12 months’ period studied, a total of 27 multidisciplinary mycology meetings were held (Table 1). In the course of these meetings, 114 patient cases were discussed, pertaining to 77 individual patients with suspected or proven fungal infection. All patients were primarily treated in Dutch hospitals, located in 6/12 of the Dutch provinces. Of these patients, 65 (84.4%) had never been discussed in the multidisciplinary meeting before the study period. Although a single consultation was sought for most patients during this period, the total number of consultations per patient varied from one to five (Table 1). Most commonly discussed aspects of care were therapy, diagnostic aspects and follow-up. In most cases, multiple aspects of patient care were discussed (64 patients, 83.1%).

Patient characteristics are provided in Table 2. Median age of adult patients (≥18 years of age) was 65 years; that of paediatric patients was 4 years at first consultation. Patients were evenly distributed among sex and all but one had a known past medical history prior to fungal infection. Most prevalent underlying diseases and comorbidities included previous pulmonary infection(s), asthma, chronic obstructive pulmonary disease, bronchiectasis, haematological malignancy and diabetes mellitus (Table 2). Of the patients with a haematological malignancy, 63.6% were children; one (adult) patient (1.3%) had undergone allogeneic stem cell transplantation. 28.6% of patients used immunosuppressive medication, including inhaled corticosteroids, and one patient was infected with (HIV).

Patients were presented at the multidisciplinary meetings primarily by a pulmonologist (37.7%), an infectious diseases specialist (35.1%) or both (3.9%), or by a paediatrician (14.3%), whereas other patients were presented by medical microbiologists, gynaecologists, otorhinolaryngologists, ophthalmologists, neurologists and clinical pharmacists.

### 3.2. Paediatric Patients

Among children (Figure 2A), 7 (53.8%) had pulmonary infections or signs/symptoms, of which 2 (28.6%) concerned IPA, 3 (42.9%) concerned pulmonary complaints or an underlying disease combined with positive fungal culture of respiratory samples of unknown significance, whereas one patient each (14.3%) had a diagnosis of invasive pulmonary fungal infection not otherwise specified (NOS) and chronic pulmonary aspergillosis (CPA). Systemic fungal infections were present in 3 (23.1%) patients: disseminated candidiasis, disseminated mucormycosis or pulmonary and central nervous system (CNS) aspergillosis were diagnosed. Furthermore, one paediatric patient each with recurrent vulvovaginal candidiasis, isolated CNS aspergillosis, and oropharyngeal and cutaneous candidiasis was discussed.

### 3.3. Adult Patients

The majority of adult patients (Figure 2B) were diagnosed with pulmonary infections or signs/symptoms (n = 40, 62.5%). Otorhinolaryngeal infections and/or fungal oesophagitis were present in 7 (10.9%) patients, pertaining to *Aspergillus* skull base osteomyelitis in 2 and *Aspergillus* sinusitis/osteomyelitis, fungal sinusitis NOS, *Aspergillus* otomycosis with possible osteomyelitis, *Candida* oesophagitis with laryngitis and possible oropharyngeal candidiasis with oesophagitis in one patient each. Systemic infections were present in 6 patients (9.4%), with three suffering from *Candida* infection (candidaemia with arthritis, candidaemia and candiduria and *Candida* arthritis with possibly infected aortic vascular prosthesis) and a diagnosis of disseminated mucormycosis, disseminated histoplasmosis and invasive pulmonary and CNS aspergillosis in one patient each. CNS infections in 6 patients (9.4%) mostly concerned cryptococcal meningitis (5 patients). Fungal skin and soft tissue infections were present in 3 (4.7%) patients, *Aspergillus* endophthalmitis in one (1.6%) and one patient was ultimately not diagnosed with a fungal infection, but with chronic granulomatous disease and disseminated nocardiosis.

The majority of adult patients with pulmonary infections or signs/symptoms (Figure 3) had a presumptive diagnosis of CPA or subacute invasive aspergillosis (SAIA) alone (19/40 patients, 47.5%). Whereas 3 patients suffered from IPA and one from allergic bronchopulmonary aspergillosis (ABPA) alone, 8 patients appeared to have a clinical overlap between CPA and ABPA, CPA and IPA or ABPA and IPA (Figure 3). This brought the total number of suspected CPA/SAIA patients to 26 and of suspected ABPA patients to 8. Furthermore, in 8 patients who displayed pulmonary signs/symptoms, radiological abnormalities and/or suffered from underlying pulmonary diseases without a clear mycological diagnosis, the significance of positive fungal microbiological or *Aspergillus* IgG test results was discussed during the multidisciplinary meeting.

### 3.4. Chronic Pulmonary Aspergillosis and Allergic Bronchopulmonary Aspergillosis

Diagnostic criteria for CPA according to the European Society of Clinical Microbiology and Infectious Diseases/European Respiratory Society (ESCMID/ERS) guidelines [18] and diagnostic criteria for ABPA according to the International Society for Human & Animal Mycology (ISHAM) guidelines [19] were retrospectively applied to the 28 patients with a presumptive diagnosis of CPA, ABPA, or both. Ultimately, 21 of these patients (75%) fulfilled the criteria for CPA and 7 (25%) fulfilled those for ABPA (one of which was not clinically suspected of ABPA, but of CPA), whereas 4 (14.3%) fulfilled criteria for neither diagnosis. More specifically, 15/28 (53.6%) fulfilled the criteria for CPA alone, 4 (14.3%) fulfilled criteria for both CPA and ABPA and 2 (7.1%) fulfilled those for CPA but were provided with insufficient data to apply ABPA criteria. Concerning ABPA, 2 (7.1%) patients could be classified as ABPA alone, and one (3.6%) as suffering from SAIA and ABPA. Of the 4 patients who did not meet CPA and/or ABPA diagnostic criteria, one (3.6%) was diagnosed with SAIA. The most commonly observed CPA subtype in patients fulfilling the ESCMID/ERS diagnostic criteria was chronic cavitating or chronic fibrosing CPA (with or without any of the other subtypes), affecting 14/21 (66.7%), whereas 5 (23.8%) suffered from *Aspergillus* noduli and one (4.8%) had a simple aspergilloma. CPA in 2 patients (9.5%) could not be categorized into a subtype.

The most common underlying diseases and comorbidities among the 21 patients with CPA according to the ESCMID/ERS diagnostic criteria were previous (probable) pulmonary infections in 12 (57.1%), asthma in 8 (38.1%), pulmonary surgery for any reason in 7 (33.3%), COPD in 5 (23.8%), bronchiectasis in 4 (19.0%), lung carcinoma with or without systemic treatment in 3 (14.3%) and pneumothorax with or without pleurodesis in 3 (14.3%). Furthermore, 7 patients (33.3%) used immunosuppressive medication (including inhaled corticosteroids), 2 suffered from sarcoidosis (9.5%), 2 (9.5%) suffered from an autoimmune disease and 2 (9.5%) had a primary immune deficiency other than cystic fibrosis (CF). One patient (4.8%) had suffered from pulmonary TBC, whereas another one had a single sputum sample positive for auramine staining and TBC PCR at the time of diagnosis, which became negative a day later (possibly reflecting contamination). None of the patients had a history of non-tuberculous mycobacterial (NTM) infection.

### 3.5. Fungal Microorganisms

A fungal pathogen was demonstrated by culture or PCR in 63/77 patients (81.8%). Among these, a single fungal pathogen was demonstrated in 50 (79.4%), whereas multiple pathogens were demonstrated in 13 (20.6%). The vast majority of these microorganisms concerned *Aspergillus* species (in 42 patients: a single species in 37 and two species in 5). *Candida* species were isolated in 9 patients (a single species in 6 and two species in 3), *Cryptococcus neoformans* was demonstrated in 6 and *Mucorales* were demonstrated in 4 patients (Figure 4).

Distribution of *Aspergillus* species was as follows: *A. fumigatus* in 28/42 (66.7%), *A. niger* in 4 (9.5%), *A. flavus* in 3 (7.1%), *A. nidulans* in 2 (4.8%), *A. terreus* in 2 (4.8%), *A. glaucus* group in one (2.4%) and *Aspergillus* NOS in 8 (19.1%). Information on voriconazole susceptibility of *A. fumigatus* isolates obtained at any time during the disease course was available for 25/28 patients in which this fungal pathogen was demonstrated. Voriconazole-resistant *Aspergillus* (minimum inhibitory concentration of ≥ 2 mg/L) was present in at least one isolate in 10/25 (40%) of patients tested. Most commonly involved *Candida* species were *C. albicans* (in 6/9 patients; 66.7%) and *C. krusei* (3 patients; 33.3%), with *C. glabrata*, *C. tropicalis* and *C. dubliniensis* being demonstrated in one patient each (11.1%). Lastly, in 2 (50%) patients suffering from *Mucorales* infection, *Lichtheimia corymbifera* was demonstrated and *Rhizomucor pussilus* was demonstrated in the other 2.

### 3.6. Antifungal Treatment

Antifungal treatment had already been administered to 56 patients (72.7%) at the time of first consultation (including the 12 patients with first consultation before 01-07-2017), and 47 patients (61%) were still on antifungal treatment. Recommendations regarding modification of antifungal treatment were given in 44 patients (57.1%). This advice pertained to switching antifungal treatment regimen (17 patients, 38.6%), starting antifungal treatment (13 patients, 29.5%), adding (5 patients, 11.4%) or removing (2 patients, 4.5%) antifungal drugs to the treatment regimen, stopping antifungals (6 patients, 13.6%), and in one patient (2.3%) switching antifungal regimen and adding an antifungal drug at a later time point.

## 4. Discussion

Our data represent an overview of the clinical cases presented at the multidisciplinary mycology meetings during a 12 months’ period in our Center of Expertise in Mycology in the Netherlands. During this period, 114 consultations were performed for 77 patients with fungal infections, most of whom were adults with a past medical history. The majority of both adult and paediatric patients discussed suffered from pulmonary infections or signs/symptoms (53.8%, and 62.5%, respectively). In the majority of cases, consultation had an (potential) impact on patients’ treatment.

Interestingly, more than half of adult patients with pulmonary infections or signs/symptoms had a presumptive clinical diagnosis of CPA or SAIA, CPA combined with IPA or CPA combined with ABPA. Initial presumptive diagnosis demonstrated relatively good agreement with formal diagnostic criteria (21/26 suspected patients fulfilled the ESCMID/ERS diagnostic criteria). The finding that most confirmed CPA patients suffer from the CCPA or CFPA subtype is in line with previous studies [20]. The high number of CPA patients discussed at the mycology meetings might be due to several factors. First, it may reflect an until recently under-recognized prevalence of this chronic infection [2]. Second, it may represent selection bias in our patient population: physicians will only discuss patients in the meetings when diagnosis, treatment and/or follow-up are problematic or when they have little or no experience with a disease entity. Awareness among clinicians regarding CPA may still be low. On the other hand, diagnosis of CPA may be difficult, especially considering the unfamiliarity of many clinicians with *Aspergillus* serological tests, a key parameter in CPA diagnosis, and the requirement of excluding alternative diagnoses. Furthermore, optimal treatment strategies and commonly accepted treatment endpoints for this disease still warrant further research, even though clinical guidelines have been published [18,21].

Underlying diseases/comorbidities in our CPA patient population are largely in agreement with those described in a large UK study [20], but differ from those observed in a recent study from Spain [22]. However, the most notable differences pertain to the presence of (previous) pulmonary TBC (4.8% in our study versus 16.7% in the UK and 18.9% in the Spanish study) and NTM infections (0 versus 14.3% and 0%, respectively) as an underlying disease. Since our hospital also hosts a centre of expertise in mycobacterial infections with its own multidisciplinary meeting, consultations for patients with mycobacterial disease might not have taken place at our mycology meetings, possibly introducing a selection bias in our patient cohort.

The relative paucity of IPA patients discussed might reflect a better acquaintance of physicians with this manifestation of pulmonary aspergillosis and the availability of well-established international and national (Dutch) treatment guidelines [21,23,24]. Furthermore, for high-risk specialties such as haematology and intensive care medicine, local treatment protocols and frequent standard consultation rounds with medical microbiologists/mycologists are in place, reducing the need for consultations at the multidisciplinary meeting. Similar reasons might apply to the relatively low number of patients discussed with *Candida* infections, despite this yeast constituting the fourth most common cause of hospital-acquired bloodstream infections in the USA [25]: mostly, cases discussed pertained to recurrent, persisting or disseminated infections, or atypical or complex presentations.

Further indications of the complexity or atypical nature of cases discussed are the finding that multiple aspects of care were discussed in 83.1% of cases and that host factor criteria for probable IFD, according to the revised European Organization for Research and Treatment of Cancer and the Mycoses Study Group Education and Research Consortium (EORTC/MSGERC) definitions [26] were present in relatively few patients. However, not all fungal infections encountered are considered invasive; allogeneic stem cell transplantation in 1.3%, immunosuppressive medication (including inhalation corticosteroids) in 28.6% and primary immunodeficiency in 10.4%; a history of recent neutropenia could not be evaluated in our cohort. The issue that EORTC/MSGERC criteria are not always applicable in all patients has been reported before [27,28] and in these cases, a multidisciplinary meeting could aid in establishing a diagnosis.

The finding that 23.1% of all paediatric and 12.5% of all adult patients discussed concerned those with respiratory signs/symptoms or an underlying disease with positive microbiological test results of unknown significance stresses the need for better diagnostic tools that help differentiate fungal disease from colonization.

The observation that 40% of patients with demonstrated *A. fumigatus* harboured a voriconazole-resistant isolate at any one time during their disease course is remarkable in light of the 10.5% of culture-positive patients with triazole-resistant isolates described in recent Dutch national surveillance data, attesting to the complexity of the patient population involved [29].

Inherent to the setup of the multidisciplinary meeting, our study does not represent a comprehensive epidemiological overview of fungal infections in the Netherlands, such as recently published [30]. Instead, it highlights the real-life difficulties encountered in daily clinical practice by a diverse array of medical specialties caring for patients with (I)FDs, revealing which questions and unmet medical needs warrant further research most pressingly in a developed country.

By combining complementary expertise, employing a structured approach and providing comprehensive advice on diagnostic work-up and antifungal treatment, the multidisciplinary mycology meeting can have a significant impact on healthcare for patients with (I)FDs. The method of practice-based registration improves the quality of consultation and provides a platform for training and education. Furthermore, obtaining experience in (I)FDs facilitates development of expert opinion-based approaches to diagnosis and treatment, potentially serving as a rationale for future clinical trials.

## 5. Conclusions

In conclusion, a multidisciplinary mycology meeting proves to be a useful modality to address complex, rare and/or atypical (I)FDs. By providing an (inter)national consultation service and thus concentrating specialized mycological care, clinical experience can be gained by the members of the multidisciplinary team in fungal diseases normally rarely encountered by a single healthcare professional. Furthermore, it may reveal unmet needs regarding diseases previously thought to be rare or those relatively unknown to clinicians, as is the case with CPA in our cohort, thus providing information on which disease entities require more awareness and which clinical issues and needs should be prioritized in further research efforts in a national or regional context.

## Figures and Tables

**Figure 1 jof-06-00274-f001:**
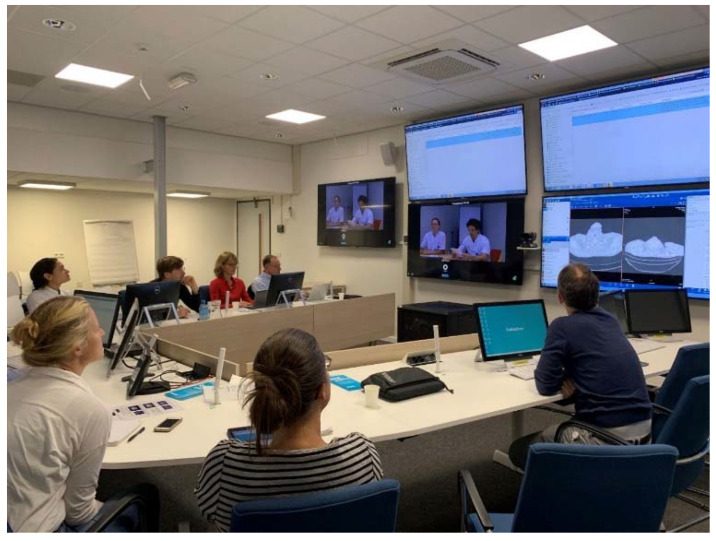
Impression of the multidisciplinary mycology meeting.

**Figure 2 jof-06-00274-f002:**
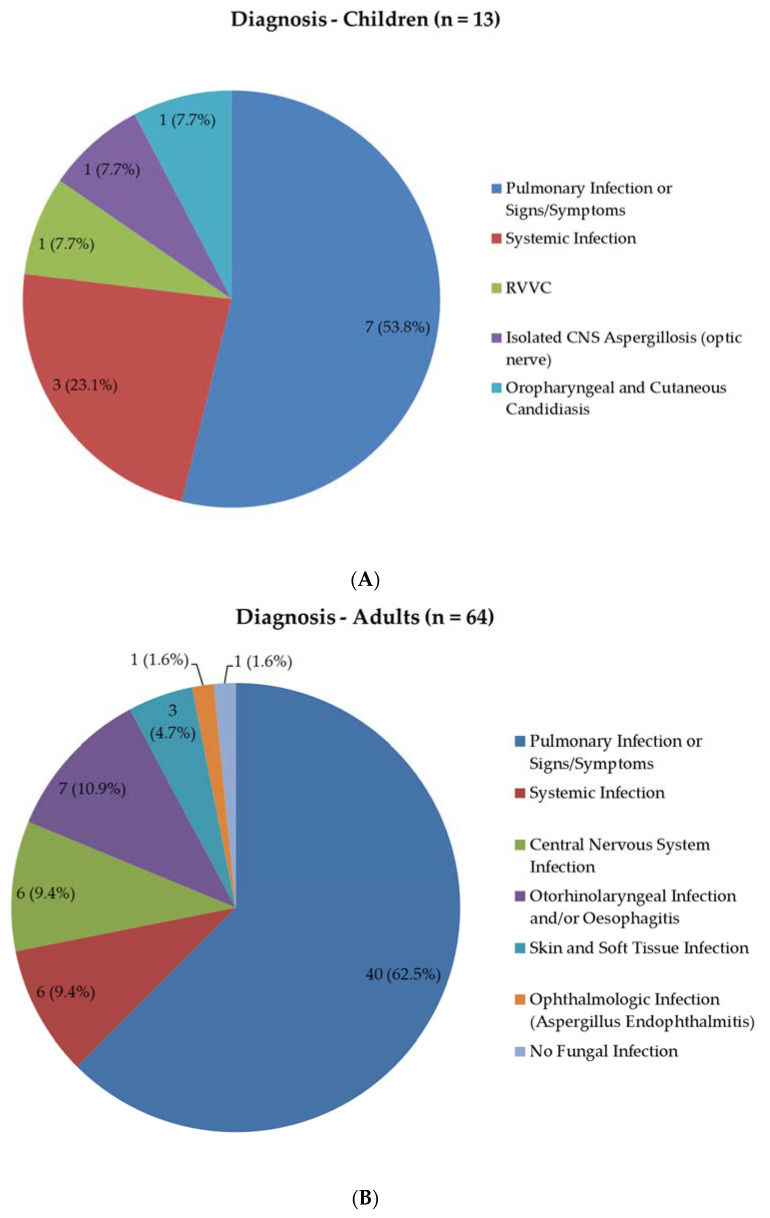
Diagnoses by organ system for paediatric (**A**) and adult (**B**) patients. RVVC, recurrent vulvovaginal candidiasis; CNS, central nervous system.

**Figure 3 jof-06-00274-f003:**
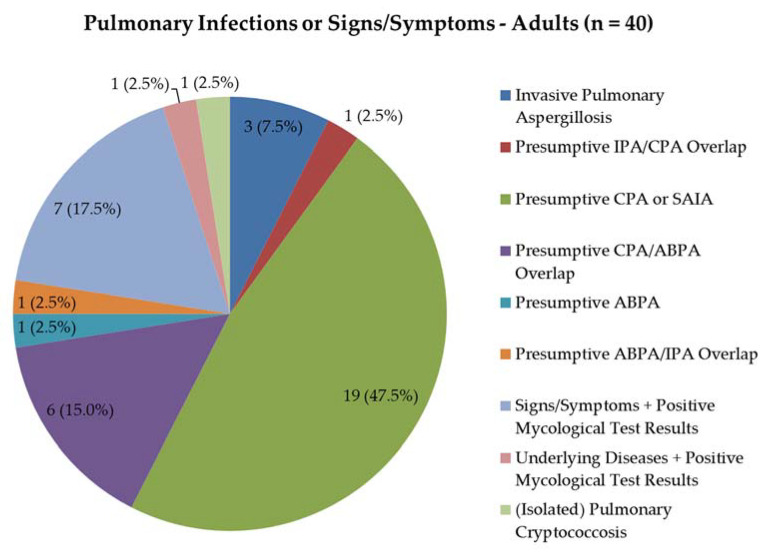
Pulmonary infections or signs/symptoms in adults. Data represent the presumptive diagnosis at the time of consultation, before retrospective application of ESCMID/ERS and ISHAM diagnostic criteria for the diagnosis of CPA, and ABPA, respectively. IPA, invasive pulmonary aspergillosis; CPA, chronic pulmonary aspergillosis; SAIA, subacute invasive aspergillosis; ABPA, allergic bronchopulmonary aspergillosis; ESCMID/ERS, European Society of Clinical Microbiology and Infectious Diseases/European Respiratory Society; ISHAM, International Society for Human & Animal Mycology.

**Figure 4 jof-06-00274-f004:**
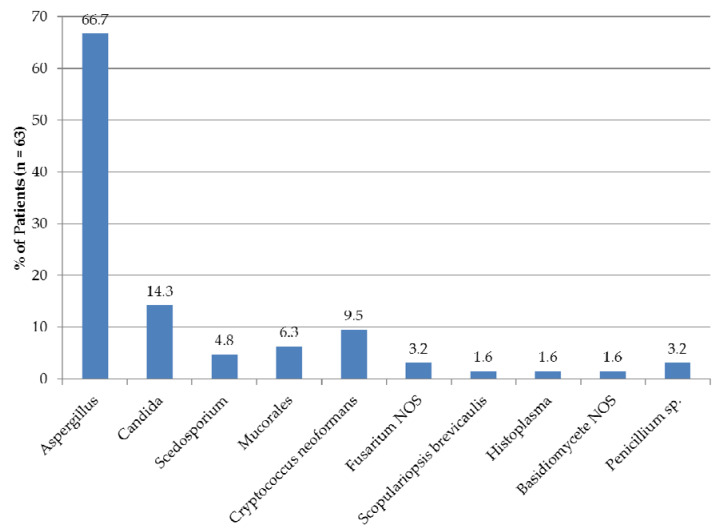
Fungal microorganisms, demonstrated by culture and/or PCR. *Penicillium* spp. and the basidiomycete NOS were not regarded as the causative microorganism in the patients involved. NOS, not otherwise specified.

**Table 1 jof-06-00274-t001:** Meeting characteristics in the period studied (1 July 2017 to 30 June 2018).

Characteristic	Value
Study period (months)	12
Number of meetings	27
Number of patient discussions (n)	114
Number of individual patients discussed (n)	77
Mean number of patients discussed per meeting (n)	4.2
Number of consultations per patient (n, %)	
1	51 (66.2)
2	20 (26)
3	3 (3.9)
4	1 (1.3)
5	2 (2.6)
Mean number of consultations per patient	1.48
Aspect(s) of care discussed (n, %) ^a^	
Single	12 (15.6)
Multiple	64 (83.1)
Therapy	70 (90.9)
Diagnostic aspects	50 (64.9)
Follow-up	38 (49.4)
Immunological aspects	17 (22.1)
Prophylaxis	4 (5.2)
Antifungal resistance	4 (5.2)
Other	1 (1.3)

^a^ Data missing for one patient (1.3%).

**Table 2 jof-06-00274-t002:** Patient characteristics (patients discussed at the multidisciplinary meeting during the study period).

Characteristic	Value
Adults (≥18 years of age; n, %)	64 (83.1)
Children (<18 years of age; n, %)	13 (16.9)
Age at first meeting (years, median, interquartile range)	
All	63.0 (37.5–69.0)
Adults (≥18 years of age)	65.0 (55.3–69.8)
Children (<18 years of age)	4.0 (2.0–15.5)
Sex (male/female; n, %)	41/36 (53.25/46.75)
Outcome at time of reporting (n, %)	
Alive	64 (83.1)
Deceased	13 (16.9)
Underlying diseases (n, %)	
Lung carcinoma +/− systemic treatment	8 (10.4)
Of which current	3 (3.9)
Other solid malignancy +/− systemic treatment	8 (10.4)
Of which current	4 (5.2)
Haematological malignancy +/− systemic treatment	11 (14.3)
Of which current	9 (11.7)
Pulmonary surgery (any reason)	9 (11.7)
Pneumothorax (+/− pleurodesis)	4 (5.2)
Previous pulmonary infections	27 (35.1)
Bronchiectasis	12 (15.6)
Cystic fibrosis	3 (3.9)
Primary immunodeficiency (other than CF)	8 (10.4)
Auto-immune disease	7 (9.1)
Solid organ transplantation	3 (3.9)
HIV/AIDS	1 (1.3)
Surgery at site of infection	7 (9.1)
Comorbidities (n, %)	
Asthma	15 (19.5)
COPD	12 (15.6)
Diabetes mellitus	10 (13.0)
Other	51 (66.2)
None	1 (1.3%)
Immunosuppressive medication (including inhalation corticosteroids)	22 (28.6)

Current: Diagnosis of disease or its systemic treatment <3 months before first symptoms or diagnosis of fungal infection; COPD, chronic obstructive pulmonary disease; CF, cystic fibrosis; HIV, human immunodeficiency virus; AIDS, acquired immunodeficiency syndrome.

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
