# Peer review of "A Multidisciplinary Approach to Fungal Infections: One-Year Experiences of a Center of Expertise in Mycology"

_jof, 2020, doi:10.3390/jof6040274_

Round 1

Reviewer 1 Report

Janssen et al reported the main issues raised during multidisciplinary mycology meetings that have been set up at the EMRC.

The paper is well written with a large amount of information.

I have few comments:

  • tab 2. the tab reports past medical history with many different clinical diagnosis or medical procedures. I think would be useful to differentiate between underlying diseases (i.e. lung carcinoma, solid malignancy, hematological malignancy....) and comorbidities (asthma, COPD....).

  • The second point I think should be stressed in the discussion is the extremely high rate of azole resistance (40%). It could be of interest to know whether there is any correlation with the type and length of prophylaxis. Based on this finding, did the physicians involved in this study modify their approach in terms of antifungal prophylaxis?

Author Response

Reviewer 1

We thank reviewer 1 for his or her constructive review of the manuscript. Please find our point-by-point responses to the reviewer’s comments below. Modifications have been made in order to address these comments and have been implemented in the revised manuscript. They have been highlighted in red (using Microsoft Word’s ‘Track Changes” function) and will be mentioned specifically below:

  • Table 2. the tab reports past medical history with many different clinical diagnosis or medical procedures. I think would be useful to differentiate between underlying diseases (i.e. lung carcinoma, solid malignancy, hematological malignancy....) and comorbidities (asthma, COPD….).

We thank the reviewer or this valuable comment. Indeed, we agree that differentiating between underlying diseases and comorbidities provides additional insights into the patient population presented at our multidisciplinary meeting. However, this distinction does provide some difficulties as to how to define an underlying illness: for some cases, this is very straightforward, as in the case with recent chemotherapy or stem cell transplantation with the occurrence of an invasive fungal infection shortly thereafter. For other cases, however, this might not be so clear: diabetes mellitus might (indirectly) contribute to the occurrence of mucormycosis, whereas a haematological malignancy treated in the far past might not contribute to the aetiology of a current fungal infection (anymore). For this reason, we chose to use the somewhat generic expression “Past medical history” to express all underlying illnesses and comorbidities. However, because we agree that especially for malignancies and their systemic treatments, the distinction between past and current disease is of importance, we have added extra information to Table 2 and distinguished “Underlying diseases” from “Comorbidities” in the Table and text (line 131)

  • The second point I think should be stressed in the discussion is the extremely high rate of azole resistance (40%). It could be of interest to know whether there is any correlation with the type and length of prophylaxis. Based on this finding, did the physicians involved in this study modify their approach in terms of antifungal prophylaxis

We thank the reviewer for pointing out this finding: indeed, voriconazole resistance present in at least one Aspergillus fumigatus isolate in 40% of patients in whom this was tested is remarkable and in our opinion might reflect a reason for presenting these aforementioned patients at our multidisciplinary meetings. However, in these cases, antifungal treatment, rather than prophylaxis, was usually the issue discussed. In the case of voriconazole resistance, 8/10 (80%) patients had been pretreated with antifungal medication before first consultation, as opposed to 11/15 (73.3%) patients in whom no voriconazole-resistant A. fumigatus isolates had been reported. Additionally, in 2/8 (25%) patients with azole resistant isolates and pretreatment before first consultation, resistance had been demonstrated before antifungal treatment had been initiated.

In case of antifungal resistance, we modify antifungal treatment based on the susceptibility patterns of the isolates involved. In case of prophylaxis, we tend not to change the antifungal prophylactic regimen based on susceptibility patterns in cases of general prophylactic regimes, based on our national or regional guidelines. In cases of prophylaxis in special circumstances (such as, but not limited to, antifungal prophylaxis in chronic granulomatous disease) that are beyond the scope of these aforementioned guidelines, we do adjust the regime accordingly. We hope this addresses the question raised by reviewer 1 sufficiently.

Reviewer 2 Report

The authors present an innovative concept for improving patient care in invasive fungal infections and introduce the overall organization as well as first demographic results of the patients discussed during the meetings. This is potentially an important add on to local AMS initiatives and makes elegant use of the tools offered by digitalization in an attempt to improve patient care. Some suggestions for further improvement:

  1. The abstract should be more condensed and focused on the major findings. E.g. “Most patients discussed were adults (XX%) and suffered from pulmonary fungal infections (XX%)” rather than resenting all the exact numbers. Furthermore, as there is currently no scientific evidence for the conclusion that these meetings “aid in the care of IFI patients” this should be rephrased.
  2. Introduction: In my opinion, the introduction could benefit from adding some information on the established value for antimicrobial stewardship and the fact that fungal infections are often neglected in AMS programs. Reasons for this could be briefly discussed and may well include lack of expertise in IFI in many AMS teams. In my feeling may be more important for the study than the rather broad epidemiological overview given in the current version.
  3. Avoid the outdated term “Zygomycetes” (Fig. 4) and use “Mucorales” (same also in the text), explain NOS (“not otherwise specified”) in Fig. 4. legend – the explanation has somehow dislocated into the text body.
  4. Please give information on the geographic origin of the cases discussed: Which geographic area is “covered” by the centers’ advice?
  5. How did the program integrate with local AMS or ID consultation programs? Clearly one of the major challenges is that such an external consultation may be perceived as an “intrusion” in local ID policies – has this ever been observed? Are local responsibles (e.g. AMS team) always invited to join the meetings?
  6. From a scientific point of view, I miss evidence that consultation in the center has really improved patient care (see also 1.). However, I acknowledge that this is a difficult-to-prove claim and as such surely not a prerequisite for publishing this initial concept description. My suggestion (rather for future studies that for the current report) would be to systematically get back to MDs who presented cases to the council and question them to monitor benefits perceived by the “customers”.

Author Response

Responses to the review comments regarding manuscript jof-983225

Reviewer 2

We thank reviewer 2 for his or her constructive review of the manuscript. Please find our point-by-point responses to the reviewer’s comments below. Modifications have been made in order to address these comments and have been implemented in the revised manuscript. They have been highlighted in red (using Microsoft Word’s ‘Track Changes” function) and will be mentioned specifically below:

  • The abstract should be more condensed and focused on the major findings. E.g. “Most patients discussed were adults (XX%) and suffered from pulmonary fungal infections (XX%)” rather than resenting all the exact numbers. Furthermore, as there is currently no scientific evidence for the conclusion that these meetings “aid in the care of IFI patients” this should be rephrased.

We thank the reviewer for the helpful comments on the abstract. We have revised the abstract and have adjusted the conclusion at the end (lines 40-53). We feel that in doing so, the abstract has been significantly improved.

  • Introduction: In my opinion, the introduction could benefit from adding some information on the established value for antimicrobial stewardship and the fact that fungal infections are often neglected in AMS programs. Reasons for this could be briefly discussed and may well include lack of expertise in IFI in many AMS teams. In my feeling may be more important for the study than the rather broad epidemiological overview given in the current version.

We thank the reviewer for addressing this important point; however, the Center of Expertise in Mycology Radboudumc/CWZ and its multidisciplinary mycology meeting currently do not deploy any antifungal stewardship activities; this would require a different setup and logistics. Instead, it provides an individualized antifungal treatment advice on a case-by-case basis. For this reason, no mention was made of antifungal stewardship in the introduction of our initial manuscript. To clarify this, we have added this information to the introduction section, while also referencing to the recently published core recommendations for antifungal stewardship (Melissa D. Johnson et al., Core Recommendations for Antifungal Stewardship: A Statement of the Mycoses Study Group Education and Research Consortium. J Infect Dis. 2020; 222 (S3): S175-98).

Therefore, no interference with the AMS program in the Radboudumc is experienced. With regard to the ID consultation program, the ID consultants are invited to join the multidisciplinary meetings and regularly do so. Furthermore, the ID physicians register patients for consultation on a regular basis.

  • Avoid the outdated term “Zygomycetes” (Fig. 4) and use “Mucorales” (same also in the text), explain NOS (“not otherwise specified”) in Fig. 4. legend – the explanation has somehow dislocated into the text body.

We thank the reviewer for pointing out these issues. We have adjusted the text (lines 230 and 243) and Figure 4. accordingly. Furthermore, the figure legend has been adjusted (lines 233-234).

  • Please give information on the geographic origin of the cases discussed: Which geographic area is “covered” by the centers’ advice

During the period described in the current study, all patients were primarily treated by hospitals in the Netherlands, based in 6/12 of the Dutch provinces. Since then, the number of hospitals consulting the multidisciplinary meeting has increased and occasionally, international consultations are performed. We have provided this geographic information on the study period in lines 120-121.

  • How did the program integrate with local AMS or ID consultation programs? Clearly one of the major challenges is that such an external consultation may be perceived as an “intrusion” in local ID policies – has this ever been observed? Are local responsibles (e.g. AMS team) always invited to join the meetings?

Please refer to our response to point 2., where we address these issues.

  • From a scientific point of view, I miss evidence that consultation in the center has really improved patient care (see also 1.). However, I acknowledge that this is a difficult-to-prove claim and as such surely not a prerequisite for publishing this initial concept description. My suggestion (rather for future studies that for the current report) would be to systematically get back to MDs who presented cases to the council and question them to monitor benefits perceived by the “customers”.

We completely agree with this point made by the reviewer. Indeed, we attempt to receive feedback on the further clinical course and outcomes of patients discussed. However, it would be of great interest to do this systematically and assess the impact of the consultations at the multidisciplinary mycology meetings on patient care.